# GLIS3: A Critical Transcription Factor in Islet β-Cell Generation

**DOI:** 10.3390/cells10123471

**Published:** 2021-12-09

**Authors:** David W. Scoville, Anton M. Jetten

**Affiliations:** Cell Biology Group, Immunity, Inflammation and Disease Laboratory, National Institute of Environmental Health Science, National Institutes of Health, Durham, NC 27709, USA; david.scoville@nih.gov

**Keywords:** GLIS3, pancreas, diabetes, development

## Abstract

Understanding of pancreatic islet biology has greatly increased over the past few decades based in part on an increased understanding of the transcription factors that guide this process. One such transcription factor that has been increasingly tied to both β-cell development and the development of diabetes in humans is *GLIS3*. Genetic deletion of *GLIS3* in mice and humans induces neonatal diabetes, while single nucleotide polymorphisms (SNPs) in *GLIS3* have been associated with both Type 1 and Type 2 diabetes. As a significant progress has been made in understanding some of *GLIS3*’s roles in pancreas development and diabetes, we sought to compare current knowledge on *GLIS3* within the pancreas to that of other islet enriched transcription factors. While *GLIS3* appears to regulate similar genes and pathways to other transcription factors, its unique roles in β-cell development and maturation make it a key target for future studies and therapy.

## 1. Introduction

The pancreas serves a dual-function role within the body. Acinar cells produce and secrete enzymes involved in digestion through the pancreatic ductal network into the duodenum to aid in digestion, while pancreatic endocrine cells play a critical role in the regulation of glycemia via hormone secretion into the blood stream. Endocrine cells cluster into islets of Langerhans, and despite their critical importance, make up only 1–4% of the pancreas [1,2]. Islets are comprised primarily of β-cells, which secrete insulin in response to elevated levels of blood glucose. Insulin insufficiency can have several causes, including insulin resistance coupled with β-cell dysfunction (Type 2 diabetes), autoimmune destruction of the β-cells (Type 1 Diabetes), a variety of monogenic causes of diabetes, as well as pregnancy induced gestational diabetes. With the exception of monogenic diabetes, where β-cell or pancreatic dysfunction is linked to mutations in one particular gene [3], diabetes mostly results from a combination of genetic and environmental factors. A variety of environmental factors (e.g., diet) have been identified that play a critical role in both Type 1 diabetes [4,5,6,7,8], and Type 2 diabetes [9,10], while genetic associations have been incredibly varied, often with small effects [11]. To better understand the interaction between environmental and genetic factors, a more detailed understanding of pancreas development and function is necessary.

The identification and characterization of a variety of transcription factors, including those associated with monogenic diabetes (*GLIS3*, *PDX1*, *PTF1A*, *HNF1A*, *HNF1B*, *HNF4A*, *FOXP3*, *PAX4*, *RFX6*, *GATA4*, *GATA6*, *NGN3*, *NEUROD1*, *PAX6*, *MNX1*, *NKX2.2*) [12], have greatly contributed to our understanding of pancreatic development. Several reviews have attempted to summarize this research [13,14,15], however new research is constantly expanding our understanding of pancreatic development and the transcription factors driving it. Previous reviews have sought to summarize GLIS3’s role in diabetes, congenital hypothyroidism, as well as a variety of other diseases [16,17,18]. Here, we sought to specifically review our current understanding of the *GLIS3* gene as it relates to pancreas development, comparing as well as contrasting its role with that of other transcription factors prominent in the field. We hope to highlight that while mice and humans lacking a functional copy of the *GLIS3* gene display many phenotypes similar to other transcription factor knockouts, the timing and features of the phenotypes differ in subtle but distinguishable ways from other transcription factors, highlighting *GLIS3*’s unique role in pancreatic development.

## 2. The *GLIS3* Gene and Its Encoded Protein

The mouse *Glis3* gene was first identified in 2003 as a gene with 5 C_2_H_2_-type zinc finger motifs that contain high homology to the *Gli* and *Zic* family of genes [19]. In humans, *GLIS3* includes 11 exons, and encodes for a protein of 930 amino acids (Figure 1A). The GLIS3 protein contains 3 known domains: an N-terminal Repressive Domain (NRD), a DNA-binding Domain (DBD) made up of the above-mentioned zinc finger motifs, and a C-terminal Transactivation Domain (TAD) (Figure 1B). The N-terminal repressive domain is largely conserved with the *GLI* family of proteins and contains amino acids that interact with the Suppressor of Fused (SUFU) protein [20]. The HECT E3 ubiquitin ligase ITCH can bind near the NRD domain in the N-terminus and promotes the polyubiquitination and degradation of GLIS3 [21]. Additionally, GLIS3 is SUMOylated on either side of the NRD by PIASy and UBC9, which inhibits its ability to stimulate transcription [22].

Despite these known interactions, there is still a significant gap in our knowledge about GLIS3 protein functions. For instance, while the C-terminal region of GLIS3 has been shown to stimulate its transcriptional activity, the proteins which interact with this domain remain undiscovered. Likewise, the GLIS3 protein is highly phosphorylated, in both N- and C-terminal domains of the protein, yet the role of these phosphorylation sites remains a mystery. Finally, GLIS3 shares very high homology within its DNA binding domains with the GLIS1 and GLIS2, allowing them to bind to similar if not identical DNA sequences. *Glis3* is also frequently co-expressed in several cell types with *Glis1* and *Glis2*. Although their DNA binding domains are highly conserved and their expression patterns regularly overlap, the phenotype of *Glis3* knockout mice is quite distinct of that of *Glis1* and *Glis2* knockout mice. This leads to unanswered questions of how GLIS3 protein is specifically recruited to the promoter and regulatory regions of its target genes and how the distinct binding of these proteins to target genes is coordinated.

## 3. Early Characterization of the *GLIS3* Knockout Mice and Humans

Examination of *Glis3* expression in mouse development revealed that it was first expressed in the early notochord, followed by expression in various neural progenitor cells [19]. *Glis3* mRNA was also detected in a variety of adult tissues, including brain, thymus, lung, kidney, uterus, skeletal muscle, pancreas, liver, and ovary. Global knockout mice were generated similarly by three different labs [23,24,25], which all reported similar pancreatic phenotypes: Hyperglycemia within the first week of life, reduced insulin expression, and early lethality (presumably due to hyperglycemia). This early characterization established that, while *Glis3* is expressed in a variety of tissues, it is likely playing a central role in the development of pancreatic β-cells.

Simultaneously, while characterization of *Glis3*’s role in the mouse provided some phenotypic information, other studies have been published showing that humans with deletions in the *GLIS3* gene also had very similar phenotypes. Affected individuals were identified as suffering from permanent neonatal diabetes and congenital hypothyroidism (NDH), as well as facial dysmorphology [26]. Some of the affected individuals also suffered from congenital glaucoma, hepatic fibrosis, and polycystic kidneys [27]. Genetic sequencing of these individuals identified various frame shift and point mutations, and deletions in *GLIS3* that were likely responsible for the observed phenotypes [27]. Additional studies have since reinforced this linkage [28,29,30,31,32,33], firmly connecting *GLIS3* to diabetes, hypothyroidism, polycystic kidney disease, as well as a host of additional phenotypes. *GLIS3*’s connection to diabetes has further been reinforced by genome wide association studies (GWAS), which we have previously reviewed [16,18]. Single Nucleotide Polymorphisms (SNPs) within *GLIS3* have been linked to both Type 1 and Type 2 diabetes, as well as gestational diabetes and decreased β-cell function. Interestingly, *GLIS3* is one of only a small number of genes that have been linked to both Type 1 and Type 2 diabetes. Taken together, these studies highlight the importance of understanding *GLIS3*’s role in pancreas development, and how it may differ from other genes linked to diabetes.

## 4. Early Pancreas Development and *Glis3* Expression in Mice

Pancreas development begins with the outgrowth of the foregut endoderm into a dorsal and ventral pancreatic bud around embryonic day 9.5 (e9.5) in mice [34,35]. Pancreas development is generally divided into two stages: a primary transition, during which time the pancreatic epithelium proliferates and undergoes extensive branching resulting in the generation of tip and trunk cells, and the secondary transition, in which via distinct differentiation pathways the three main lineages that make up the mature pancreas are generated [36]. The trunk domain is made up of bipotent progenitor cells, which differentiate into ductal and endocrine cells, and tip domain cells, which primarily form acinar cells, although this domain also contains multipotent progenitor cells capable of producing all pancreatic cell types [36]. While a few glucagon+ and insulin+ cells are observed during the primary transition [37], endocrine cell development primarily occurs during the secondary transition, starting at around e13.5 [38]. Endocrine progenitor cells de-laminate from the bipotent trunk domain, then differentiate into the five different cell types that comprise the islet: α-, β-, δ-, ε-, and pp-cells.

Differentiation of the pancreas has largely been characterized by the stepwise expression of a variety of transcription factors (for full reviews, see [14,35,39,40]). Early dorsal and ventral pancreatic buds are marked by expression of the transcription factor *Pdx1* and *Ptf1a* [41,42,43]. Deletion of *Pdx1* results in pancreatic agenesis in mice [41,44], and a similar phenotype was observed in human patients with mutations in *PDX1* [45,46,47,48,49]. Similarly, *Ptf1a* has early expression in the pancreas beginning at e9.5 and is a marker of the multipotent progenitor population within the pancreas, while also playing a role in the acinar cells [50,51]. Deletion of *Ptf1a* also results in pancreatic agenesis similar to *Pdx1* [43]. These two factors mark some of the earliest markers of pancreatic development, and as one might expect, their deletion is detrimental to the early formation of the pancreas.

Additional transcription factors are also expressed early in the primary transition, and like *Pdx1* and *Ptf1a*, their expression changes as the pancreas forms its branched structure of tip and trunk domains [15]. For example, *Sox9* is first expressed at low levels at e10.5 in multi-potent progenitor cells (MPCs) [52], and its expression remains high in bipotent progenitors prior to being restricted to ductal cells. *Hnf6* and *Hnf1b* both follow similar patterns of expression in the bipotent progenitor cells, followed by restriction to the ductal lineage later in development, while *Foxa2* is expressed earlier than *Hnf6* and *Hnf1b* and is maintained in all 3 lineages. As may be expected based on their expression pattern, deletion of *Sox9*, *Hnf6*, or *Hnf1b* results in a form of pancreatic hypoplasia, or a general loss of pancreatic cells [53,54,55,56,57]. Correspondingly, deletions of *HNF1B* result in a similar phenotype in humans [58]. Deletion of *Foxa2* produces even earlier disruption in notochord development [59,60,61].

*Glis3* mRNA expression was first detected at e11.5 in both the dorsal and ventral pancreas [24], but expression of GLIS3 protein was only detected starting at e13.5 [62], indicating it likely does not play a role in early pancreas development. Indeed, analysis of *Glis3* global knockout mice failed to detect any effect on overall pancreas morphology or acinar cell development [24], consistent with its lack of expression in early pancreatic progenitor cells.

## 5. *Glis3*’s Role in the Secondary Transition and Ductal/Endocrine Lineage Determinations

The secondary transition marks a period of differentiation for acinar, ductal, and endocrine cells. While acinar cells differentiate from the “tip” region of the developing pancreatic branches, the trunk domain is composed of bipotent progenitor cells that can differentiate into either ductal or endocrine lineages. Endocrine progenitor cells are distinguished primarily by their high, transient expression of *Ngn3*, whereas ductal progenitors express *Hes1*. A form of lateral inhibition has been suggested to drive these fate decisions, as *Ngn3* has been linked to upregulation of Notch signaling, which in turn upregulates *Hes1*, which itself inhibits *Ngn3* [63,64]. This is likely only one mechanism involved in making the ductal/endocrine decision, as many HES1^+^ cells have been observed lacking neighboring NGN3^+^ cells.

PDX1^+^, SOX9^+^, NKX6.1^+^ bipotent progenitor cells represent the first stage where GLIS3 protein could be detected in a GFP knockin mouse expressing a fusion GLIS3-GFP protein [62]. *Glis3* expression is maintained in subsequent differentiation into both the productal (HNF6^+^, SOX9^+^) and proendocrine (NGN3^+^) cells (Figure 2). This distinguishes *Glis3* from many of the other transcription factors expressed during this period. While many are expressed during the bipotent progenitor stage, expression is often limited to either the ductal or endocrine lineage. This expression restriction presumably helps drive the differentiation process, as knockouts for many of these factors leads to impairment of the subsequent cell type differentiation. Interestingly, in *Glis3* knockout mice, the endocrine lineage is dramatically affected, whereas the ductal lineage does not appear to be affected prior to duct formation [24]. This indicates that *Glis3* likely does not play a role in lineage decision-making in the bipotent progenitor cells, but instead plays important roles during or after allocation to the ductal or endocrine lineages.

As mentioned, bipotent progenitor cells commit to the endocrine lineage through their expression of *Ngn3*, *Isl1*, and *Neurod1*, as well as several other factors. *Ngn3* is a transient marker of endocrine progenitor cells whose expression is mostly lost in mature endocrine cells [38,65], although there is evidence that some *Ngn3* expression is required for postnatal β-cells [66]. *Isl1* and *Neurod1* expression is maintained during the differentiation of endocrine progenitors into α-, β-, δ-, and pp-cells [67,68]. Interestingly, *Neurod1* is also expressed earlier in the small number of glucagon^+^ cells present in the primary transition [68], although its function in these cells is unclear. Consistent with transcription factors expressed during the primary transition, transcription factors expressed predominantly in the endocrine lineage during the secondary transition play a critical role in the regulation of endocrine cell differentiation. Deletion of *Ngn3* or *Neurod1* results in similar phenotypes, with postnatal pancreas lacking endocrine cells and mice dying of hyperglycemia due to a lack of insulin [65,68]. Pancreas-specific deletion of *Isl1* produces a similar phenotype [69], although global *Isl1* knockouts die earlier due to heart defects.

The phenotype of *Glis3* mice is therefore most similar to that of other transcription factors controlling the endocrine lineage. *Glis3* global knockouts display decreased gene expression and staining for all endocrine hormones, and pups die within the first 10 days, likely due to hyperglycemia [23,24,25]. One of the genes that is decreased in *Glis3* knockout embryonic pancreas is *Ngn3*, providing a potential mechanism for the decrease in endocrine cell number. Conditional knockout of *Glis3* using a pancreas specific (*Pdx1-cre*) produced a different phenotype, with islets present well after birth, and α/δ- cells appearing relatively unaffected [70]. This could potentially be due to a slightly later deletion of *Glis3* allowing for more *Ngn3* expression during development. Alternatively, as the *Pdx1-cre* line is known to be mosaic (deletion efficiency ranged from about 50% to about 80%), it is possible that a small but significant number of GLIS3^+^ cells persisted during embryonic development, allowing for the establishment of a sufficient endocrine progenitor population. A more in-depth analysis of GLIS3’s role during endocrine development could help distinguish what role it plays in guiding this process.

## 6. Links between *Glis3* and β-Cell Maturation

Following embryonic and early postnatal development, β-cells within pancreatic islets undergo a still-poorly defined process known as maturation [71]. This occurs roughly around weaning in mice, when pups transition from a primarily milk-fat based diet to a carbohydrate diet. Mature islets not only secrete more insulin in response to glucose, but more tightly regulate their insulin secretion [72]. This is due to a variety of changes in the metabolism of β-cells (reviewed in [71]). In mice, the expression of two transcription factors, *Mafa* and *Mafb* [73], is often used for marking β-cell maturation, where *Mafb* expression is repressed, while *Mafa* is exclusively expressed in mature pancreatic β-cells.

*Glis3*’s expression is maintained in pancreatic β-cells, from the immature to mature state [62]. Similarly, *Mafa* is expressed starting at around e13.5 in insulin^+^ cells and its expression is maintained in mature β-cells [73]. *Mafa* expression appears to be directly regulated by *Glis3* within pancreatic islets [70], in line with the observed decreased expression of *Mafa* in β-cells from *Glis3* knockout mice, and rising hyperglycemia. *Glis3* also directly regulates *Ins2* expression, which is significantly down across all *Glis3* knockout models [23,24,25,70]. This correlates well with what is seen in humans (discussed in the next section), where *GLIS3* appears to be a critical regulator of *INS* expression [74].

The phenotypes of *Mafa* and *Mafb* knockout mice are different from that of the transcription factors mentioned in the previous section (*Isl1*, *Neurod1*, and *Ngn3*). *Mafa* knockout mice produce insulin^+^ β-cells but exhibit impaired insulin secretion in response to glucose challenge as early as three to four weeks postnatally [75,76,77]. While islet β-cell mass is modestly reduced in these mice, defects in glucose tolerance are thought to be driven primarily by defects in insulin secretion [76,77]. Pancreas-specific *Mafb* knockout mice, unlike *Mafa* knockout mice, appear to have normal glucose clearance in 3-week glucose tolerance tests [78]. In addition, unlike *Mafa* knockout mice, *Mafb* knockout mice exhibit an embryonic phenotype, with reduced numbers of α- and β-cells during prenatal development. *Mafa*/*Mafb* double knockouts die shortly after birth, presumably due to hyperglycemia from a lack of islet β-cells [78]. *Glis3* potentially plays a role in directly regulating both *Mafa* and *Mafb* expression, as it binds to a presumptive enhancer and promoter region, respectively, and their expression is downregulated in *Glis3* pancreas-specific knockouts [70]. Thus, *Glis3* may act upstream of both *Mafa* and *Mafb* during β-cell development and affect β-cell maturation via their regulation.

Research over the past decade has suggested that not only is the activation of many genes required for β-cell differentiation and maturation, but also the repression of certain genes. These latter genes have been termed “disallowed” genes, in that their downregulation is correlated with β-cell identity and function [79,80]. These genes include *Acot7*, *Cox5a*, *Fam59a*, *Gas6*, *Itih5*, *Ldha*, *Lmo4*, *Mgst1*, *Nfib*, *Pdgfra*, *Plec1*, *Rpl36*, *Tgm2*, *Tst*, and *Zdhhc9*, which are upregulated in Type 2 diabetic islets [80]. Other studies have identified a de-differentiation pathway in response to extreme cellular stress, which involves the upregulation of genes expressed primarily during β-cell development and subsequently silenced, such as *Ngn3*, *Oct4*, *Nanog*, and *L-myc* [81]. Of note, none of these genes were upregulated in *Glis3* knockout mice, highlighting the unique role that *Glis3* is likely playing in guiding β-cell maturation and function [70].

## 7. *GLIS3* in Human β-Cell Development

Pancreas development in the human seems to largely follow a similar pattern to that of mice, with some key differences [82]. One key difference between mice and humans is the apparent lack of a first wave of INS^+^ or GCG^+^ cells observed early in mouse pancreas development [83]. This is possibly due to subtle delays in human pancreas development compared to mice, preventing early differentiation of endocrine cells. Additionally, human endocrine progenitor cells lack the expression of *NKX2.2*, a transcription factor that is critical for beta cell development in mice [84]. Unfortunately, due to a lack of available antibodies, we do not know the expression of *GLIS3* during human pancreatic development. It is therefore possible that *GLIS3* expression and function in human pancreatic development differs somewhat from that of its role in mouse described above, although humans with deletions in *GLIS3* develop neonatal diabetes similar to mice due to a lack of insulin (reviewed in [16]).

A useful tool in studying human pancreatic development has been the differentiation of human embryonic stem cells (hESCs). A significant amount of research has been devoted to the differentiation of human pancreatic β-cells from hESCs, with the hope of developing a potential therapy for people with Type 1 Diabetes [85]. While this differentiation does not exactly mimic human pancreatic development, many of the differentiation stages obtained do express the appropriate marker genes similar to what has been observed in humans. The advent of CRISPR technology and its use in hESCs allows for the study of many of the pancreatic genes identified in mice to be studied in humans [86,87]. Disruption of the *GLIS3* gene function by deletions within the DNA binding domain revealed that, in the absence of a functional copy of *GLIS3*, hESCs were able to differentiate into a similar number of PDX1^+^ and C-PEPTIDE^+^ cells as normal hESCs [87]. Defects were observed in *PDX1*, *RFX6*, and *NGN3* disrupted hESCs, suggesting that in humans *GLIS3* may play a later role in human β-cell development than in mice. Interestingly, a subsequent study using a CRISPR knockout of *GLIS3* function saw a reduction in INS^+^ cells, as well as reductions in the expression of several critical transcription factors, such as *PDX1*, *MAFA*, *NKX6.1*, and *NEUROD1* [88]. The authors attributed this phenotype to an increase in cell death due to activation of the TGFβ pathway, a finding not previously observed in mice [70], but supported by a study using cell lines [89].

Of note, the differentiation protocols initially used by Zhu et al. showed minimal upregulation of *GLIS3*, although their protocol produced largely INS^+^GCG^+^ cells [87]. The subsequent differentiation protocol used by Amin et al. produced significantly more INS^+^GCG^−^ cells and saw a greater increase in *GLIS3* expression [88]. This could suggest that, in humans, *GLIS3* functions as a crucial regulator at a relatively later stage of β-cell differentiation or maturation. Additionally, GLIS3 protein seems to coordinate this function with other transcription factors via co-binding of genomic loci, as has been reported for ISL1 and LDB1 [70,90]. Similar results have been observed in human islets, with GLIS3 binding to the human *INS* promoter region together with other transcription factors [74]. Not only does GLIS3 appear to bind and activate the *INS* promoter, but it appears to be required for binding of PDX1 and NEUROD1 to the *INS* promoter as well [74]. This evidence further supports a model of β-cell transcription factor co-binding and coordination where multiple transcription factors bind to overlapping regions and are required for proper gene regulation.

Studies in a rat insulinoma cell line (INS-1E cells) and dissociated human islets have indicated that *GLIS3* may play a role in preventing β-cell apoptosis, including in response to cytokine treatment [89]. The proposed mechanism of action involves *GLIS3* regulation of *SRSF6* (also known as *SRP55*), a splicing factor that is down-regulated in human islets upon cytokine treatment [91], and is involved in the splicing of a variety of critical genes in a human β-cell line [92]. By regulating the alternative splicing of *Bim*, down-regulation of SRSF6 promotes apoptosis in β-cells by increasing the generation of the proapoptotic isoform BIM S [93,94,95]. Alternatively, *GLIS3* has also been proposed to regulate apoptosis through the TGF-β pathway during differentiation of hESCs to β-like cells [88]. Unfortunately, these studies offer confusing and sometimes conflicting results for GLIS3 function in humans, as we have previously highlighted [16]. GLIS3 regulation of *SRSF6* does not appear to be transcriptional, as hESCs and *Glis3* knockout mice show normal expression of *SRSF6* mRNA [70,88]. This highlights one of the difficulties of the human models that currently exist: Apoptosis is dramatically higher in cell lines and in vitro culture systems than has been observed in vivo.

A separate study also linked *Glis3* to regulation of the *Manf* gene, an anti-apoptotic gene upregulated in β-cells during unfolded protein stress response (UPR) [96]. However, this study did not examine apoptosis in their model (*Glis3* heterozygous mice undergoing β-cell specific UPR), but instead relied on previous reports from cell lines [89]. *Manf* was not downregulated in *Glis3* knockout mice, but GLIS3 protein does appear to bind to a regulatory region within the mouse *Manf* gene, alongside other islet-enriched transcription factors [70]. This raises the possibility that GLIS3 regulation of certain genes may be context specific and be dependent on signaling pathways activated by certain conditions, such as cellular stress or cytokine exposure. Clearly, more work is needed to address whether *GLIS3* is linked to apoptosis in humans or mice in vivo and under what circumstances.

## 8. Newly Identified Human Mutations within *GLIS3*

*GLIS3* has been linked to both Type 1 and Type 2 diabetes, gestational diabetes, and β-cell function in many GWAS (previously reviewed in [16]). Many of the SNPs identified in these studies reside within the first few introns of the *GLIS3* gene, likely regulating *GLIS3* expression through enhancer elements. Additional reports identified a link between several deletions and mutations within GLIS3 and neonatal diabetes with congenital hypothyroidism (NDH) syndrome (reviewed in [18]). Recently, novel mutations within the coding region of *GLIS3* gene have been identified. One such novel homozygous mutation created a premature stop codon within the C-terminus of *GLIS3* (c.2392C>T; p.Gln798Ter) and caused a syndrome characterized by neonatal diabetes, congenital hypothyroidism, congenital glaucoma, and cystic kidney disease [97]. The premature stop codon lies within the known transactivation of the protein (see Figure 1B), thus confirming the functional conservation of the domain from mice [98]. Separately, a novel heterozygous mutation was identified in the N-terminal region (c.589G>T; Asp197Tyr) of *GLIS3*, in a Turkish patient diagnosed with maturity onset diabetes of the young (MODY) [99]. However, this N-terminal mutation (Asp197Tyr) lies outside any of the previously identified domains of GLIS3. This mutation could therefore provide interesting insights into the previously unknown functional domains within GLIS3. Hopefully, as genetic sequencing of patients increases in both frequency and thoroughness, novel mutations could allow us not only to extend our knowledge of GLIS3 function in humans but may also point us toward which regions of the gene to explore in mice and in vitro human models.

## 9. Conclusions

A significant body of research now exists tying GLIS3 to the regulation of pancreatic β-cells and diabetes. *GLIS3* deficiency significantly reduces the generation of endocrine cells, particularly β-cells, causing severe hyperglycemia in both mice and humans. *G*LIS3’s expression pattern and phenotype within the pancreas of *Glis3* knockout mice most closely resembles that of transcription factors that play a role in endocrine differentiation, such as NGN3, ISL1, and NEUROD1. However, GLIS3 appears to play a distinct role from these factors in that, while total endocrine cell numbers are reduced, β- and γ-cell numbers are more severely affected in *Glis3* knockout mice. Moreover, GLIS3 appears to be necessary for the binding of additional transcription factors to the human *INS* promoter [74]. And *GLIS3* is one of only a few genes that have been linked through GWAS to both Type 1 and Type 2 diabetes.

Significant questions remain as to the exact role of GLIS3 in the human β-cells and how closely it mimics the function in mice. For instance, *Ngn3* expression has been reported to be downregulated in global *Glis3* knockout mice at e13.5 and e15.5 [24,25]. GLIS3 binding does overlap with that of PDX1 in a region downstream from the *Ngn3* gene consistent with the concept that it is directly regulated by GLIS3 [25]. However, the differentiation of human ESCs into β-cells in which *GLIS3* was deleted appeared to have no effect, whereas *NGN3* deletion has a dramatic effect on β-cell generation. This lack of a G*LIS3* phenotype might be attributed to the in vitro and artificial nature of the protocol failing to recapitulate human development or to inherent differences in GLIS3 function between human and mouse. Additional studies using primary human islets might establish a more physiologically relevant assessment of *GLIS3* function and provide greater insights into the potential of GLIS3 as a therapeutic target in the management of diabetes.

## Figures and Tables

**Figure 1 cells-10-03471-f001:**
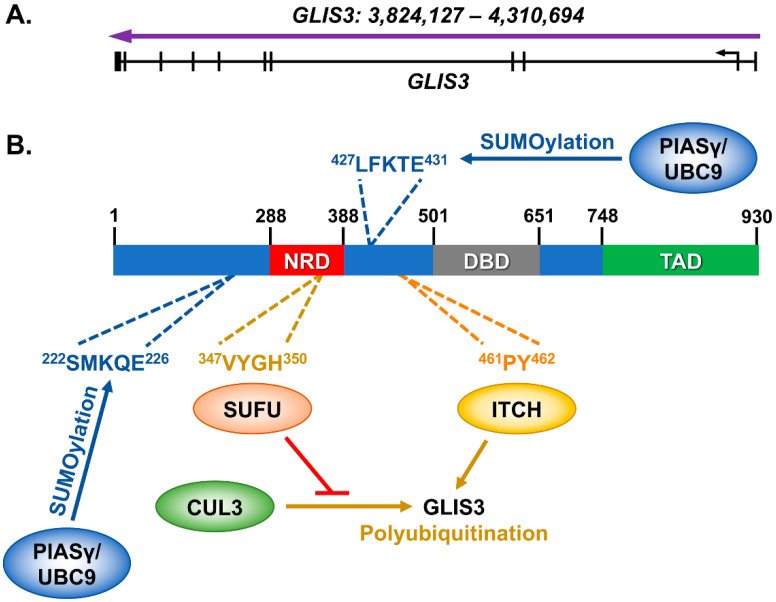
GLIS3 and its known interacting proteins. (**A**) The *GLIS3* gene (NM_001042413) is composed of 11 exons, which produces (**B**) a 930 amino acid protein. Interactions have previously been identified with SUFU [20], ITCH [21], and PIASy/UBC9 [22] with either mouse or human GLIS3 protein. SUFU interaction inhibits GLIS3 polyubiquitination by the E3 ubiquitin ligase CUL3. The GLIS3 amino acids that are interacted with or modified are indicated.

**Figure 2 cells-10-03471-f002:**
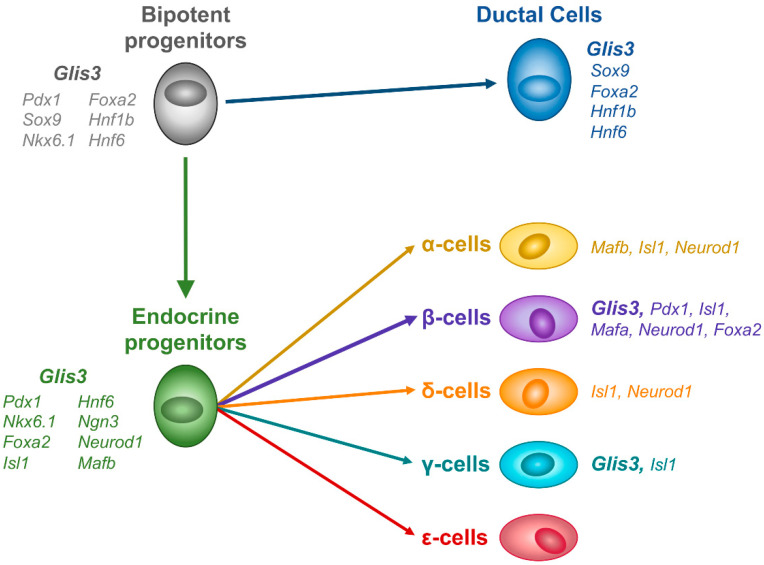
GLIS3 expression during mouse endocrine development. GLIS3 protein is first detected in bipotent progenitor cells, and remains expressed in both ductal and endocrine lineages, before becoming restricted to β- and γ-cells.

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
