# Peer review of "GLIS3: A Critical Transcription Factor in Islet β-Cell Generation"

_cells, 2021, doi:10.3390/cells10123471_

Round 1

Reviewer 1 Report

The manuscript titled “GLIS3: A Critical Transcription Factor in Islet β-cell Generation” by Scoville et.al. highlights the importance of the GLIS3 transcription factor during early pancreas development and beta-cell differentiation. This manuscript also summarizes the interaction of GLIS3 with other islet transcription factors during beta-cell development.

Comments:

Most of the data present link the role of GLIS3 to development and beta-cell maturation in mice. However, confounding data from human cells have considerably dampened the significance of this topic.

Albeit, the authors effectively discussed the discrepancy in the role of GLIS3 between the mice and human counterparts.

Stem cell-based beta-cell developmental models although seem like the closest thing to natural development, differences in the differentiation protocols may create conflicting results, therefore, the use of human primary islets to understand and verify the findings may be imminent.

Overall, I congratulate the authors for a thorough summary of the GLIS3 transcription factor and its role in beta-cell development and function.

Author Response

Herewith we are submitting our revised manuscript entitled “GLIS3: A Critical Transcription Factor in Islet β-cell Generation”. We are pleased that our manuscript was received well and appreciate the comments and suggestions made by the reviewers that have led to an improved manuscript. 

Rebuttal to reviewer’s comments:

Reviewer 1:

Comment 1: Stem cell-based beta-cell developmental models although seem like the closest thing to natural development, differences in the differentiation protocols may create conflicting results, therefore, the use of human primary islets to understand and verify the findings may be imminent.

Response: We agree with the reviewer and have included a sentence highlighting the importance of human primary islets in future studies (lines 367-369).

Reviewer 2 Report

  1. Why are only some transcription factors that are associated with monogenic diabetes listed? (line 38)
  2. Please include UCN3 if authors are focusing on markers for beta-cell maturation and include its relevance to GLIS3. On the other hand, if authors are focusing only on beta-cell maturation markers that act as transcription factors, please change the text accordingly (line 209-239)
  3. Authors should highlight the expression of MAFA during embryonic development and its selective re-expression in adult mouse beta cells
  4. The focus of this review was to highlight how GLIS3 acts in a unique way compared to other transcription factors during beta-cell development and maturation. This is missing in the conclusions section (line 339-357)
  5. Please include more details about each of the mutations associated with GLIS3 that are included in this article
  6. Highlight the importance of this article over the other review articles on the similar topic

Minor correction:

  1. Provide expansion of GWAS when it is used for the first time (line 102)
  2. Please maintain italics for Hnf6 and Hnf1b (line 140-141)
  3. Possible typo in line 294 for INS1-E

Author Response

Herewith we are submitting our revised manuscript entitled “GLIS3: A Critical Transcription Factor in Islet β-cell Generation”. We are pleased that our manuscript was received well and appreciate the comments and suggestions made by the reviewers that have led to an improved manuscript.

Reviewer 2:

Comment 1: Why are only some transcription factors that are associated with monogenic diabetes listed? (line 38).

Response: We have updated this list to include all currently known transcription factors leading to monogenic diabetes.

Comment 2: Please include UCN3 if authors are focusing on markers for beta-cell maturation and include its relevance to GLIS3. On the other hand, if authors are focusing only on beta-cell maturation markers that act as transcription factors, please change the text accordingly (lines 209-239).

Response: Our review focuses on the role of transcription factor in pancreatic b cells. We have altered the text accordingly to highlight this more clearly.

Comment 3: Authors should highlight the expression of MAFA during embryonic development and its selective re-expression in adult mouse beta cells.

Response: We have expanded on the role of Mafa gene to clarify the timing of the expression of MAFA in the text (lines 221-222).

Comment 4: The focus of this review was to highlight how GLIS3 acts in a unique way compared to other transcription factors during beta-cell development and maturation. This is missing in the conclusions section (line 339-357).

Response: We agree with the reviewer this was not well described. The first part of the conclusion section has been revised to explain the similarities and differences between GLIS3 and other transcription factors more clearly. In the second part of the conclusions, we discuss some of the remaining questions in the field.

Comment 5: Please include more details about each of the mutations associated with GLIS3 that are included in this article.

Response: Mutations and variants were recently detailed in reference 16. In this review, we only focused on the newly identified mutants. We changed the heading of this section to make this clearer (line 325) and provided more detail on the newly discovered mutants (lines 335-340).

Comment 6: Highlight the importance of this article over the other review articles on the similar topic.

Response: We have revised the introduction to highlight the objective of this review and the difference between this review and other reviews examining GLIS3 (lines 43-47).